# Motivation biases behavior but not perception
Christian Wolf [1] ✉, Markus Lappe [1] & Hugh Riddell [2,3]

Why do people differ in their perceptual judgment despite observing the same situation? According to "motivated perception", a person's motivation can alter how the brain interprets incoming sensory information. Yet, empirical support remains mixed, often due to methodological confounds. Here, we systematically tested whether motivation alters perception or whether it instead biases behavior. Across four experiments, we assessed the quality and quantity of motivation (self-concordance, value) and two key dimensions of perception: bias and sensitivity. Moreover, we tested two potentially mediating mechanisms (gaze position, spatial attention) as well as an implicit perceptual measure. Using smooth pursuit eye movements as an implicit measure of motion perception, we show that motivation biases responses without altering perception (Experiment 2, n = 20). Changes in perceptual sensitivity (Experiment 1, n = 60) and perceptual bias (Experiments 3–4, n = 16 and 24) only arise when participants can freely select their gaze position in response to uncertain or ambiguous visual displays. Our findings therefore challenge the notion of "motivated perception". Instead, they suggest that motivation shapes how we look and respond – but not how we perceive.

Imagine watching a soccer match in which a shot hits the goal line, leaving it ambiguous whether the ball fully crossed the line. According to accounts of motivated perception[1–4], observers are more likely to perceive the ball as having crossed the line when the apparent goal favors their own team. Motivated perception refers to phenomena suggesting that the process responsible for initiating and maintaining goal pursuit (i.e., motivation) influences how the brain constructs a representation of the environment from incoming sensory information (i.e., perception). Indeed, motivation has been described to change reported percepts of world events, covering vision[5–9], audition[10], and olfaction[11]. For example, Balcetis and Dunning[6] told people that the category of animal images (farm animal versus sea creature) would predict whether they would receive a pleasant or unpleasant food item after the experiment. The crucial image shown was an ambiguous seal-donkey figure, and most people reported perceiving the animal associated with the pleasant item.

Work questioning motivational effects on perception[12–16] explains these effects by demand characteristics and methodological pitfalls. A key challenge is distinguishing perception from report. When confronted with the same objective situation, differences between people may arise because their motivation alters their perception, biasing how sensory input is interpreted. Alternatively, people perceive the same thing, yet report or respond differently to align with a desired outcome. Thus, the question remains: does motivation change perception, or does it change behavior? In laboratory experiments, motivated perception is typically studied using ambiguous or near-threshold stimuli, which allow researchers to capture subtle, often unconscious biases that participants themselves might not be aware of. However, because most experiments rely on perceptual reports from participants, teasing perception and response apart can be difficult.

How then can we dissociate perception and response? One approach uses computational modeling to identify latent variables. Signal-detection theory[17] separates bias from perceptual sensitivity but cannot distinguish whether a bias reflects altered perception or response strategy. Similarly, drift-diffusion modeling[18] interprets changes in the starting point of evidence accumulation as response bias, and changes in the rate of evidence accumulation (drift rate) as perceptual[19–21]. Yet even this distinction is not clear-cut: First, changes in the drift rate can be explained by dynamically evolving response biases[22,23]. Second, changes in the drift rate can also be observed in studies investigating reward-based motor decisions[24–26] (rather than perceptual judgments), suggesting that changes in the drift rate might instead reflect changes in motivation – and not changes in perception.

A second approach relies on neural or behavioral correlates of perception. Neuroimaging can unveil perceptual processing, but is also sensitive to reporting demands[27,28]. A promising alternative involves implicit behavioral readouts – such as smooth pursuit eye movements – which reflect perceptual processing without requiring explicit judgments[27,29,30]. Because they unfold online and continuously, such measures provide a unique window into perceptual dynamics, relatively uncontaminated by post-perceptual processes.

[1]Institute for Psychology, University of Münster, Münster, Germany. [2]Curtin School of Population Health, Curtin University, Perth, WA, Australia. [3]enAble Institute, Curtin University, Perth, WA, Australia. ✉e-mail: chr.wolf@uni-muenster.de

If motivation indeed shapes perception, then their relationship might be more nuanced than biasing the interpretation of incoming sensory information. First, motivation may not impinge directly on perception, but motivational effects may be mediated by processes such as attention and gaze. Motivation can modulate visual selective attention[31], which in turn alters perceptual processing[32,33]. Similarly, in visual tasks, motivation manifests in more accurate eye movements[24,25] that in turn support perceptual fidelity[34,35]. Second, any motivational effect on perception might not necessarily manifest as a perceptual bias, but could instead affect another key perceptual dimension: sensitivity – that is, how well something is perceived rather than what is perceived. Such effects are particularly plausible considering the vast differences in visual sensitivity across the visual field[36,37] and attentional modulations thereof[32,33]. Finally, any relationship between motivation and perception might not necessarily be about motivational quantity (i.e., whether or how much somebody is motivated), but about motivational quality (i.e., why somebody is motivated). Contemporary theories such as Self-Determination Theory[38,39] emphasize the importance of motivational quality rather than quantity for initiating, sustaining, and benefiting from goal-directed behavior. Critically, internally motivated goals (commonly termed self-concordant[40]) produce more effective goal-directed behavior than externally motivated ones[41]. Despite this, most motivated perception studies to date have focused on motivational quantity only.

The aim of the current study was to put the relationship between motivation and perception to a clear and rigorous test. Across four experiments, we assessed distinct motivational facets (value, self-concordance) and two key dimensions of perception (bias, sensitivity), while simultaneously controlling for possible mediating processes (gaze position, spatial attention) and distinguishing perceptual and response biases. We hypothesized that if the notion of motivated perception was correct, motivation should directly influence perceptual sensitivity or perceptual bias. Alternatively, if the apparent relationship between motivation and perception arises from mediating processes, differences in perceptual sensitivity and bias should be explained by factors such as gaze position, spatial attention, or response biases. Our results provide strong, convergent evidence that motivational value biases behavior – response behavior and gaze behavior. Gaze behavior, in turn, explains differences in perceptual sensitivity, or the perceptual biases found with ambiguous stimuli.

## Methods

### Transparency and openness
We report all data exclusions, manipulations, and measurements, and justify sample sizes. Hypotheses, dependent variables, sample size justification, trial number, and analysis plan of Experiment 1 were preregistered (https://doi.org/10.17605/OSF.IO/Q79HR) on 2023-10-19, prior to data collection (starting 2023-10-27). Deviations from the preregistration are mentioned in the "Methods" section and/or Supplementary Table 1. Experiments 2–4 were not preregistered.

### Participants
We recorded data of 60 healthy adults for Experiment 1 ($M = 22$ years, SD = 4.8; range: 18–51, 10 males, 50 females; see Supplementary Note 1). Experiment 2 was piloted with two naïve participants, yielding an effect size of $d = 6.39$ (paired $t$-test, psychophysical bias versus oculomotor bias). A power analysis using G*Power 3.1[42] indicated that three participants would be required to detect this effect (two-tailed paired $t$-test, alpha = 0.05, power = 0.95). To ensure normality and to provide robust findings, we decided to record 20 participants (age range: 20–34, 6 males, 14 females). Experiments 3 and 4 included convenience samples of 16 (age range: 20–33, 4 males, 12 females) and 24 participants (age range: 19–27, 4 males, 20 females). Participants from all experiments were undergraduate students from the University of Münster and were reimbursed with course credit or 8€/h. Additionally, participants of Experiments 1 and 2 received a performance-related payment (Experiment 1: 0.50–2.60€, median = 1.70€; Experiment 2: 6.60–9.40€, median = 8.40€).

The experimental procedure was approved by the local ethics committee (ID: 2023-47-ChW), and all experiments were conducted in accordance with this approved protocol and the Declaration of Helsinki. All participants provided written informed consent before testing. Gender was assessed using self-report. No data on ethnicity or socioeconomic status were collected.

### Motivation quality (C-RAI)
Following Self-Determination Theory[39], motivation quality was assessed using the Comprehensive Relative Autonomy Index (C-RAI)[43], a validated 24-item scale, comprising subscales of autonomous and controlled motivation. Self-concordance was calculated by taking the difference between autonomous and controlled motivation subscales. Raw scores were z-standardized for analysis.

The item stems were adapted to suit each experimental task, and participants of all experiments filled out the questionnaire after a brief demo to familiarize them with the task. We translated the scale into German using a forward-back procedure[44]. The translated subscales demonstrated good reliability in all experiments ($\alpha_{controlled} = 0.79$–0.84, $\alpha_{autonomous} = 0.81$–0.91). More details on the translation process are available in the Supplementary Methods. The translated scale is publicly available (https://doi.org/10.17605/OSF.IO/2S5V6).

### Experiment 1 (sensitivity experiment)
**Stimuli and setup.** In Experiment 1, we used four different stimulus categories (Fig. 1): (i) a pre-saccadic disc, two post-saccadic targets containing either (ii) only noise or (iii) an additional digit covered in noise, and (iv) Gabor gratings. The former three were used in eye movement trials, the latter in attention trials.

The pre-saccadic disc (2° diameter) was slightly darker (4.9 cd/m²) than the background (8.75 cd/m²). Post-saccadic targets contained luminance noise, maximal within the central 0.71° (21 pixels) and gradually fading toward the disc periphery. Noise values were Gaussian and independent for each pixel.

Half of the post-saccadic targets additionally contained one of eight digits (1, 2, 3, 4, 6, 7, 8, 9) covered in noise (Fig. 1A). Digit luminance was either brighter (~8 cd/m²) or darker (~2 cd/m²) than the disc. For digit pixels, noise strength was reduced to 20%.

Gabor gratings (1.2 cycles/deg) were seen through a Gaussian envelope (sd = 0.3°) and tilted by ±10° relative to vertical. Grating contrast was initially set to 0.82 (Michelson contrast). Two QUEST staircases[45] (one per location) subsequently controlled the Michelson contrast within a range of 0.01–0.9, using a prior contrast of 0.2 (sd = 0.3), derived from piloting. Stimuli were presented using the Psychtoolbox[46] in MATLAB (The MathWorks, Natick, MA). Eye position of the right eye was recorded at 1000 Hz using the EyeLink1000 (SR Research, Mississauga, ON, Canada) and the EyeLink Toolbox[47].

**Eye movement trials.** At trial onset, a fixation cross appeared 6° left of the screen center (Fig. 1B). After a random interval, a gray disc appeared 12° to the right. Provided that participants made a saccade that crossed the vertical monitor midline, they were free to look anywhere. Once the horizontal gaze position crossed the vertical monitor midline, the pre-saccadic disc split into the two post-saccadic disks, vertically separated by 6° (±3° from the horizontal monitor midline). Each disc contained visual noise, with or without an embedded digit, shown for 307 ms. Afterwards, one disc remained on screen (without any stimulus or noise), indicating to participants to report the digit presence/absence for this disc using the keyboard. After the participants' response, the obtained score per trial was shown for 107 ms as feedback. Correct "digit absent" responses yielded 5 points, whereas correct "digit present" responses yielded points equal to the digit's value. If incorrect, 5 points were subtracted. Collected points were transformed into a monetary reward at the end of the experiment (500 points = 1€).

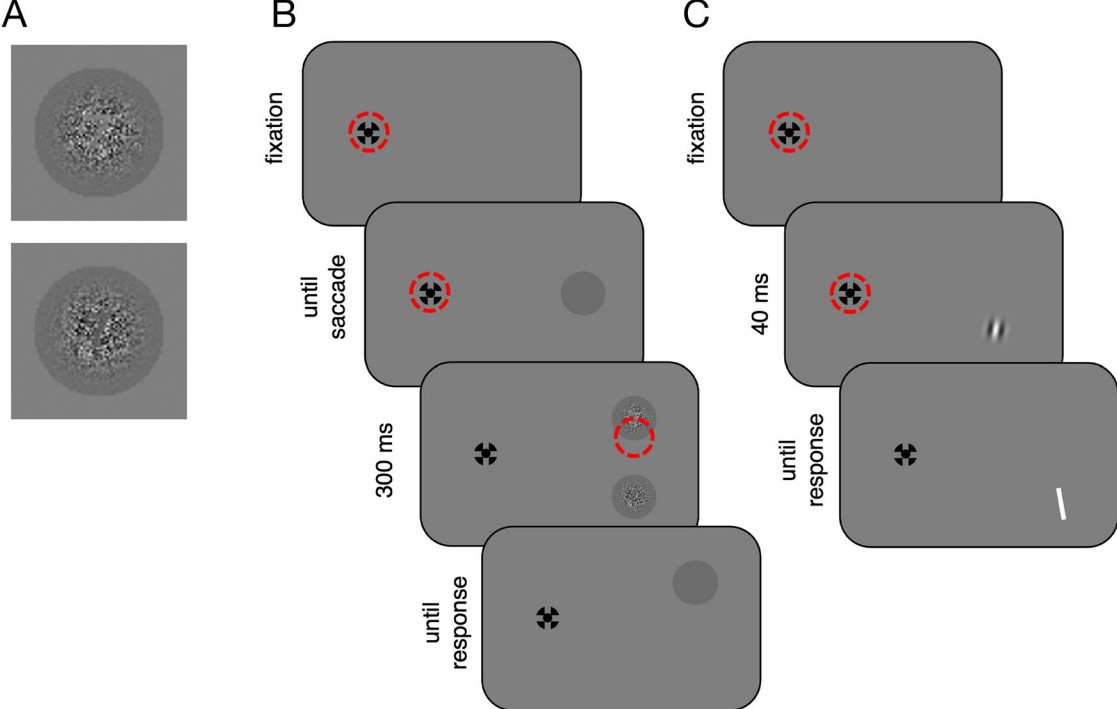

**Fig. 1 | Experiment 1: stimuli and trial procedure. A** Example stimuli used in eye movement trials. Half of the post-saccadic targets contained a digit (Helvetica font; approximately 0.5° height) embedded in noise. Digits could be either brighter (upper example) or darker (lower example) than the background disc. **B** Trial procedure in eye movement trials and **C** trial procedure in attention trials. Red dashed circles (not shown during actual experiment) denote an example gaze position on the initial fixation cross and during stimulus presentation. In attention trials, participants reported the perceived orientation of the grating by selecting one of two orientations of the white bar.

Crucially, one post-saccadic disc mostly contained high-value digits (>5), whereas the other disc mostly contained low-value digits (<5). We refer to these disks as high-value and low-value targets, respectively. The two locations were balanced across participants. Digits were randomly drawn from a normal distribution: For the high-value (low-value) target, approximately 75% (25%) of digits were above 5, with the digit 7 (3) being the most frequent digit for most participants. Digit value and digit presence were independent across disks. We made it explicit to participants that (i) one location mostly contained high-value digits, whereas the other location mostly contained low-value digits, (ii) which disc was high value and which disc was low value, and (iii) that knowing a digit was present/absent for one location did not provide any information about the other location. Hence, participants were motivated to perceive a digit at the high-value location and no digit at the low-value location. Nonetheless, they could maximize their monetary reward if they responded accurately[20,21].

The experiment comprised 500 trials (400 eye movement trials, 100 attention trials), randomly interleaved. Prior to the main experiment, participants completed a 30-trial demo (24 eye movement trials, 6 attention trials) that was easier than the actual experiment (longer stimulus presentation, higher contrast) but approached the experimental difficulty throughout the demo. The only possible score that could be obtained or lost was ±5.

**Attention trials**. During attention trials (Fig. 1C), participants fixated the fixation cross while a Gabor grating was shown at one of the two post-saccadic locations (40 ms). Afterwards, a vertical bar appeared at that location, and participants indicated whether the grating was tilted clockwise or counterclockwise. Correct and incorrect responses yielded ±5 points, respectively. There were 50 trials for each location.

**Data analysis**. Saccade onsets and offsets were defined using the Eye-Link1000 algorithm. Perceptual responses in eye movement trials were analyzed using signal-detection theory[17], yielding criterion (response bias) and d′ (perceptual sensitivity) separately for the high- and low-value target location.

In attention trials, we measured contrast thresholds for each location using the QUEST procedure[45], resulting in a posterior distribution. We used the posterior's mean as an index of contrast thresholds. Given that means were not normally distributed, we deviated from our preregistered analysis plan and computed a normalized difference score instead: the attentional imbalance. The attentional imbalance, $\Delta_{attention}$, was computed as:

$$\Delta_{attention} = \frac{(M_{low} - M_{high})}{(M_{low} + M_{high})} \tag{1}$$

Thus, $\Delta_{attention}$ values > 0 indicate higher thresholds (i.e., worse performance) at the low-value location.

Inferential statistics were carried out in JASP 3[48] and R. To test the effects of motivational value (i.e., difference between high-value and low-value location) on criterion, d′, attentional imbalance, and saccade endpoints, we used one-sided t-tests, as our preregistered hypotheses regarding motivational value were directed. To assess the relationship between two variables (non-directed hypotheses), we used linear regressions. We based our conclusions on frequentist statistics but supplemented our analyses with the equivalent Bayesian test using default priors. Bayes Factors (BF$_{10}$) > 1 favor the alternative, and values < 1 favor the null-hypothesis, with values between 0.33 and 3 typically considered inconclusive or anecdotal evidence[49].

Assumptions of normality were assessed by visually inspecting data distributions. Except for the contrast threshold measure described above, the data were approximately normally distributed.

## Experiment 2 (bias experiment)

**Stimuli, procedure, and design.** Participants pursued two successively presented moving dots and indicated which moved faster. First, a red dot (0.5° diameter) appeared at the screen center. After a random interval, the dot was displaced 1° to the left and moved rightward (step-ramp-paradigm[50]). Thereafter, a second dot appeared and moved similarly. A central "1 or 2?" signaled participants to respond. Motion durations varied (540, 720, 900 ms) to avoid judgments about final position, rather than velocity. We explicitly told participants that neither the dots' final position nor their presentation duration is informative about their velocity.

We used the method of constant stimuli: one dot, the standard stimulus, moved with 8°/s, whereas the comparison moved with one out of nine possible velocities (4, 6, 7, 7.5, 8, 8.5, 9, 10, or 12°/s). Each block contained 216 trials, consisting of 36 trials for each of the three most difficult comparisons (7.5, 8, 8.5°/s) and 18 trials for each of the remaining ones. Stimulus order, motion duration, and comparison velocity were balanced.

Participants completed two blocks. In the second (biased) block, participants received an additional response-contingent reward. Importantly, reward magnitude depended on the chosen response: Half of the participants received a large monetary reward (10 cents) when correctly indicating the first stimulus as faster, and a small reward (1 cent) for correctly indicating the second stimulus as faster (reversed for the other half). Consistent with Experiment 1 and prior studies on "motivated perception"[20,21,51], participants made the most money if they responded accurately. Participants completed eight unbiased practice trials of random velocities before the experiment.

**Data analysis.** Eye velocity was obtained by differentiating eye position. Position and velocity signals were filtered using Butterworth filters, and saccades were removed from velocity traces using linear interpolation. Pursuit velocity was averaged 250–500 ms after target onset. To assess oculometric functions[52], we computed the proportion of trials in which comparisons were pursued faster than the standard. Psychometric/oculometric functions were fit using psignifit4[53] in MATLAB R2021b (The Mathworks, Natick, MA, USA). For each participant, we fitted four psychometric and four oculometric functions, one for each combination of stimulus order (standard first vs second) and block (unbiased vs biased). We computed two bias indices derived from points of subjective equality (PSE): the psychophysical bias and the oculomotor bias. Therefore, we computed the PSE difference from the biased block, relative to the same difference from the unbiased block. Values > 0 indicate that participants more frequently pursued (oculomotor bias) or reported (psychophysical bias) the highly rewarded stimulus. Normality was formally tested using the Shapiro–Wilk test. Bias indices were compared against 0 using one-sample $t$-tests and against each other using a paired-sample $t$-test.

## Experiments 3 and 4 (ambiguity experiments)

We conducted two experiments using ambiguous stimuli. Experiment 3 tested whether spontaneous and intentional reversals in perception are accompanied by differences in gaze position. Experiment 4 tested whether enforced differences in gaze position bias perception.

**Stimuli.** Both experiments used ambiguous visual illusions and composite images. Illusion stimuli included the seal-donkey illusion (also known as the horse-seal figure), the duck-rabbit illusion, the young-woman-old-woman illusion (also known as "My wife and my mother-in-law"), and the B-13 illusion, obtained from publicly available online resources (Google image search). Stimuli were displayed with a size of 19.4 × 19.8° (seal-donkey), 19.4 × 12.9° (duck-rabbit), 11.6 × 15.9° (young-old woman), and 11.6 × 10.56° (B-13). Composite stimuli were created by morphing face and house images. Face images were taken from the Chicago Face Database[54], house images were taken from the SUN database[55]. For Experiment 3, 30 composite stimuli were created with equal contributions of face and house information. For Experiment 4, we selected four composite stimuli and varied the relative face-house proportion (0.2, 0.33, 0.42, 0.48, 0.5, 0.52, 0.58, 0.66, 0.8). Composite images were displayed with a size of 19.4 × 19.4°.

For each stimulus used in Experiment 4, we had pre-selected two diagnostic image locations that were preferentially chosen by participants in Experiment 3. For the composite stimuli, selected house features overlapped with the face in the image.

**Procedure and design.** Experiment 3 consisted of 34 trials, each comprising three phases. At the beginning of each trial, text indicated the two possible percepts (e.g., "seal versus donkey"). In Phase 1 (spontaneous perception), participants freely viewed the stimulus and continuously reported the currently dominant percept via button press. This phase captured spontaneous perceptual reversals and ended when participants pressed the space bar. In Phases 2 and 3 (intentional bias), participants were instructed to bias their perception toward one of the two possible percepts. Once successful, participants pressed the space bar and were asked to maintain their percept for the remaining 2500 ms of stimulus presentation. The trial order was randomized across participants.

Experiment 4 tested whether enforced fixations on diagnostic image features bias perception. It comprised two blocks: an illusion block (40 trials) and a composite block (560 trials). In both blocks, stimuli were viewed while participants fixated on one of two predefined image locations. At trial onset, a fixation cross was displayed on the screen. Once participants looked at the fixation cross, the image appeared for 307 ms or once participants' gaze moved more than 1.5° away from the fixation location. The image location on the screen was jittered by introducing a random horizontal (-1, 0, +1°) and vertical offset (-1, 0, +1°).

In the illusion block, each stimulus was shown 10 times (five times per fixation location) in a random order. Participants reported which percept was dominant in their perception. The composite block contained 280 trials for each fixation location (and thus for each psychometric function). Participants indicated whether the face or the house had a higher intensity in the image. For each fixation location, the difficult intensity proportions (0.42, 0.48, 0.5, 0.52, 0.58) were repeated 40 times, and the easier proportions (0.2, 0.33, 0.66, 0.8) were repeated 20 times. The trial order was randomized.

**Analysis.** To analyze whether spontaneous reversals are preceded by systematic gaze differences, we measured the gaze distance to diagnostic image regions associated with face perception[56]. For composite images, we used the location between both eyes; for illusion stimuli, we selected the eyes of the seal, the eyes of the young lady, and the top gap in the B-13 image. Time courses of gaze distances were temporally aligned with perceptual reversals and compared using a cluster-permutation test. Therefore, we randomly permuted the assignment between individual time courses and the dominant perception using 1000 permutations. For every comparison, we report the sum of $t$-values, $t_{sum}$, as an index of the cluster strength of the original data, the critical cluster value, $t_{crit}$, the $p$-value, and the time window of the (strongest) cluster. The critical cluster value is determined by computing the strongest cluster for every permutation and then taking the 95th percentile. The $p$-value is given by the proportion of permuted clusters exceeding the cluster strength in the original data.

To analyze whether gaze positions differ when participants intentionally bias their perception in ambiguous stimuli, we used permutation tests (1000 permutations) by permuting the assignment between individual fixation locations and the intended percept. For illusion stimuli, we compared gaze positions between the two percepts. For composite stimuli, we compared the distance to the eye region when participants perceived the face or the house. For each permutation test, we report the observed gaze distance, $\Delta$gaze, the $p$-value, as well as the critical gaze distances, $\Delta$gaze$_{crit}$, thus the 95th percentile of the permuted gaze distance.

To analyze whether enforced fixations bias perception (Experiment 4), we compared perceptual reports to chance (0.5) using $t$-tests. Normality was

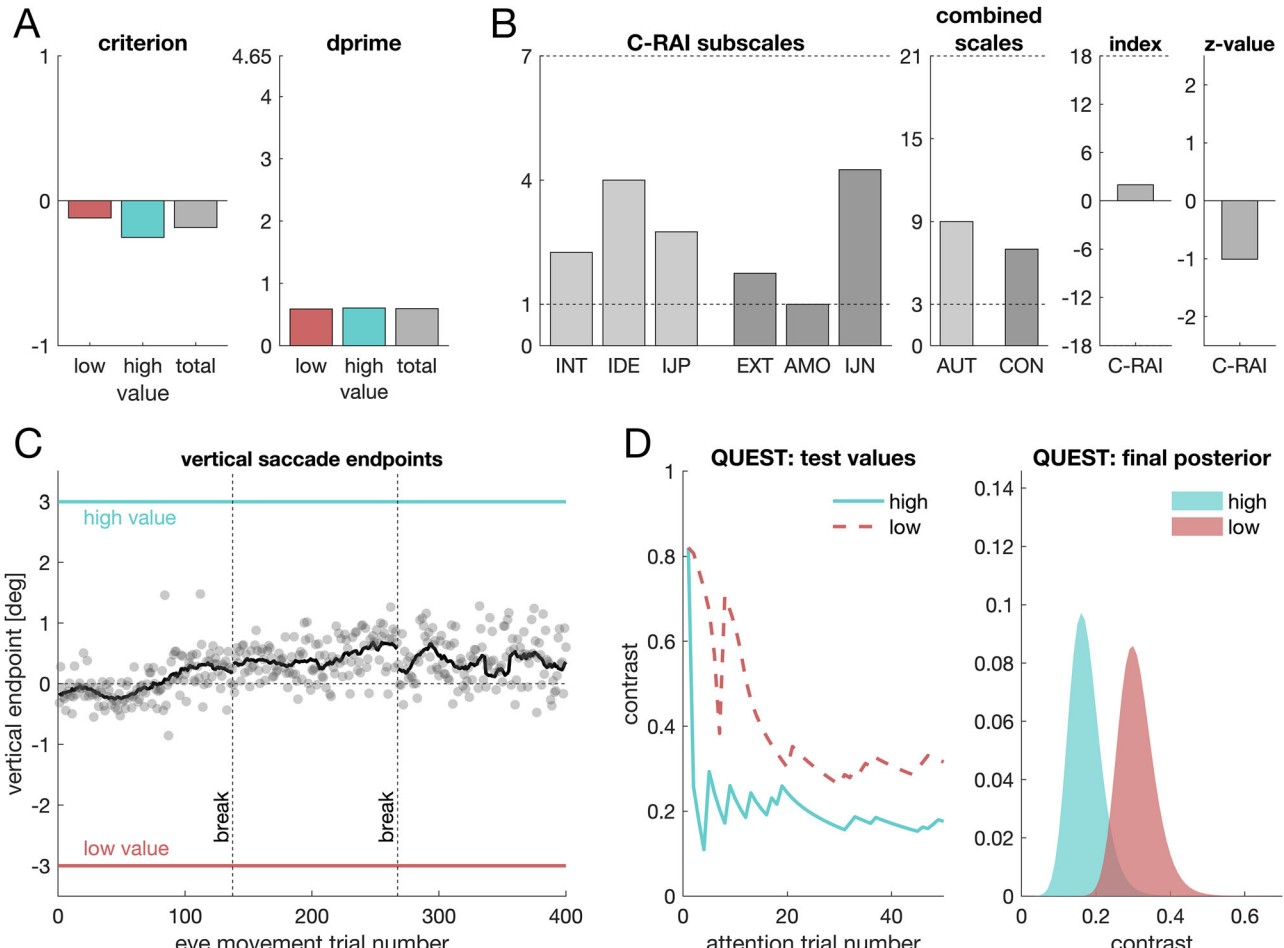

**Fig. 2 | Experiment 1: individual data of one representative participant. A** Signal-detection scores (criterion and $d'$) obtained from behavioral responses in eye movement trials ($n = 1$ participant). **B** C-RAI data (motivation quality). The left panel shows the average item score for the six subscales (INT intrinsic, IDE identified, IJP positive introjection, EXT external, AMO amotivated, IJN negative introjection) with the sum of the former three yielding the value for the combined scale autonomous (AUT), and the latter three yielding the controlled scale (CON). The lowest and the highest possible values are denoted by the horizontally dashed lines. A self-concordance index reflecting motivation quality is given by the difference between autonomous and controlled motivation. **C** Vertical saccade endpoints. Each dot represents one trial; the black line is a running average that was computed by means of a sliding Gaussian window. Colored horizontal lines denote the respective target positions. Vertically dashed lines indicate experimental breaks. **D** Left panel: Contrast values of the Gabor gratings shown in attention trials over the course of the experiment, separate for the two QUEST staircases. Right panel: Final posterior distributions.

formally tested using the Shapiro–Wilk test. Although the data deviated from normality, we report $t$-tests for consistency. Nonparametric tests (Wilcoxon signed-rank tests) yielded the same conclusion (Supplementary Table 2). For composite stimuli, we computed two psychometric functions for each observer using psignifit[53], one for each fixation condition. We compared points of subjective equality using $t$-tests.

## Results

### Motivation quantity does not affect perceptual sensitivity after controlling for gaze position (Experiment 1)

In Experiment 1, participants detected digits embedded within visual noise (Fig. 1A) – a challenging perceptual task that can best be solved by directly looking at the stimulus. We simultaneously recorded eye movements as an index of gaze strategy (Fig. 1B), and assessed contrast thresholds at each target location as an index of spatial attention (Fig. 1C). The reward structure was chosen so that participants were motivated to perceive a digit at the high-value location and no digit at the low-value location, yet accurate responses maximized their payoff [20,21].

Figure 2 shows the data from a representative participant. We used signal-detection theory to analyze response behavior (Figs. 2A, 3A, B) and to distinguish between bias (criterion) and sensitivity ($d'$). Mean $d'$ values ($M_{high} = 0.63$, $SD_{high} = 0.56$, $M_{low} = 0.45$, $SD_{low} = 0.49$) were not larger for the high- compared to the low-value location, $t(59) = 1.68$, $p = 0.050$, $d = 0.22$, 95% CI $[-0.04, 0.47]$, $BF_{10} = 0.997$ (anecdotal evidence) (Fig. 3B). Thus, the difference in perceptual sensitivity did not cross the predefined statistical threshold. Nevertheless, when sensitivity is compared without accounting for potential mediating mechanisms, descriptively higher $d'$ values are observed at the high-value location.

We therefore next analyzed spatial attention and gaze position as potential determinants of perceptual sensitivity. Attentional performance did not differ between locations, $t(59) = 0.118$, $p = 0.453$, $d = 0.02$, 95% CI $[-0.24, 0.27]$, $BF_{10} = 0.155$ (Fig. 3C), nor did we find evidence that sensitivity was related to spatial attention, $F(1, 58) = 0.7$, $p = 0.405$, $BF_M = 0.352$ (anecdotal evidence). Hence, we did not find any evidence that spatial attention mediates the effects of motivational quantity on perceptual sensitivity.

In contrast, mean saccade endpoints were biased towards the high-value location, M = 0.3° (SD = 0.98), $t(59) = 2.39$, $p = 0.010$, $d = 0.31$, 95% CI $[0.05, 0.57]$, $BF_{10} = 3.85$ (Fig. 3D). Participants with an average endpoint closer to the high-value location had higher sensitivity for that location, $F(1, 58) = 23.08$, $p < 0.001$, $\beta = 0.534$, $BF_M = 1506$ (Fig. 3E) and lower sensitivity for the low-value location, $F(1, 58) = 20.71$, $p < 0.001$, $\beta = -0.513$, $BF_M = 681.6$. Hence, perceptual sensitivity depended on gaze behavior. A within-participant analysis yielded the same conclusions (Supplementary Note 2).

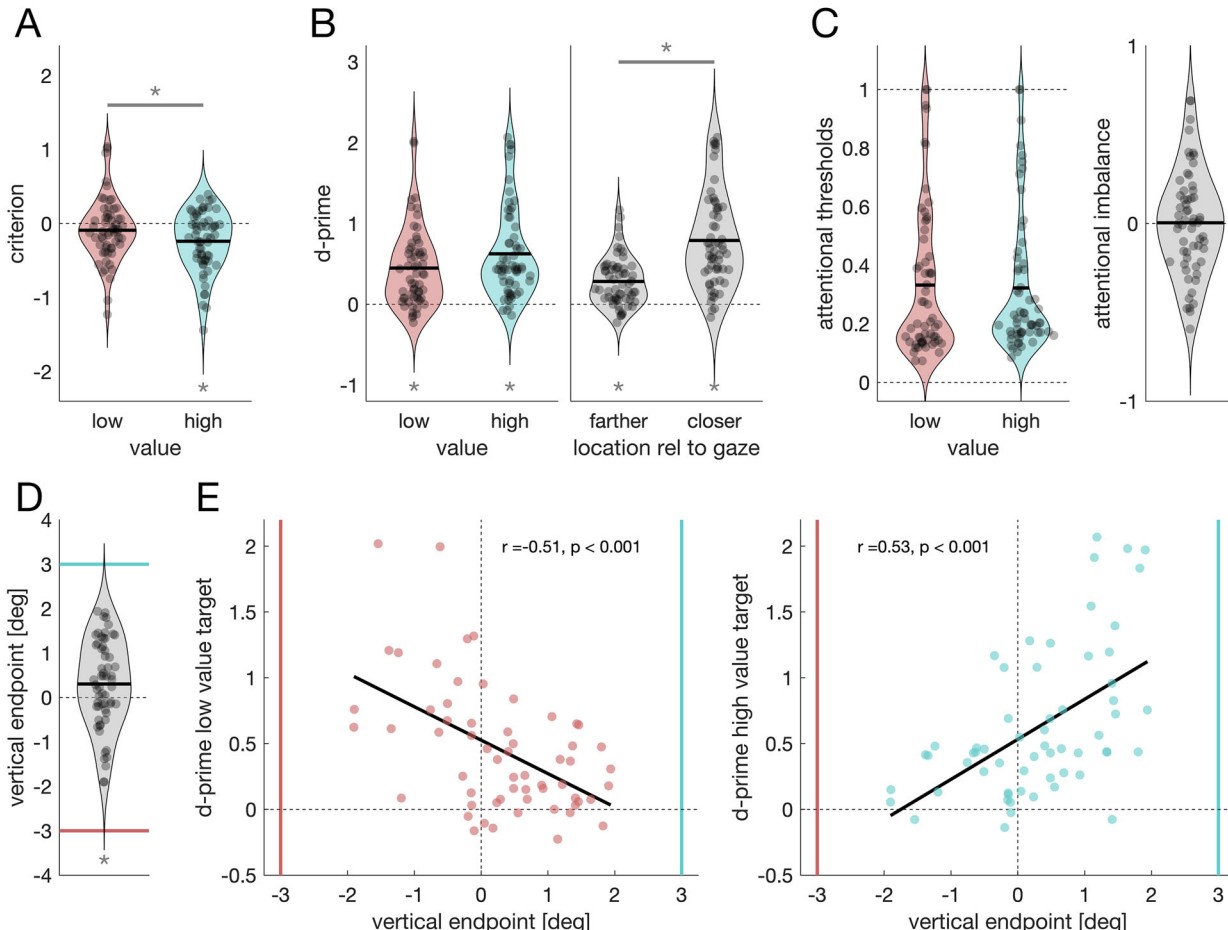

**Fig. 3 | Experiment 1: behavioral results. A** Violin plots for criterion values obtained from signal-detection theory for the low (red color) and the high-value location (turquoise color). Each dot denotes one individual (*n* = 60 participants), the solid black line indicates the overall mean. Asterisks below a distribution indicate a significant difference from 0; asterisks and lines above indicate a difference between value locations. **B** *d′* for the low- and high-value location (left panel) as well as for the location that is farther or closer relative to an individual's mean endpoint. **C** Left panel: Attentional thresholds (i.e., the posterior mean). Right panel: attentional imbalance score. **D** Mean vertical endpoint. The asterisk denotes a significant difference from 0. **E** Relationship between eye movement endpoints and *d′*, for the low-value target (left panel) and the high-value target (right panel). Colored lines in (**D**) and (**E**) indicate the respective target positions.

For both target locations, the regressions between *d′* and saccade endpoint yielded a nearly identical intercept ($\beta_0$ = 0.53; Fig. 3E), suggesting that differences in sensitivity can be explained by differences in gaze position. To test this notion, we compared *d′* values after controlling for saccade endpoints by first regressing *d′* on individual saccade endpoints and then comparing residuals for the low-value and high-value locations. We found no evidence for differences in perceptual sensitivity when gaze behavior was considered, *t*(59) = 0.081, *p* = 0.936, *d* = 0.01, 95% CI [−0.24, 0.26], $BF_{10}$ = 0.142. Importantly, differences in *d′* between the high- and low-value location were significantly reduced after controlling for gaze position, *t*(59) = 2.39, *p* = 0.020, *d* = 0.31, 95% CI [0.05, 0.57], $BF_{10}$ = 1.95 (anecdotal evidence). Taken together, this shows that any apparent motivational effect on perceptual sensitivity is mediated by gaze behavior.

**Motivation quantity biases responses (Experiment 1)**
Criterion values (Fig. 3A) were negative for the high-value location, indicating a bias towards reporting digits as present ($M_{high}$ = −0.24, $SD_{high}$ = 0.40), $t_{high}$(59) = −4.58, *p* < 0.001, *d* = 0.59, 95% CI [0.32, 0.86], $BF_{10}$ = 794.89. Criterion values for the low-value target were not significantly above zero ($M_{low}$ = −0.09, $SD_{low}$ = 0.42), $t_{low}$(59) = −1.65, *p* = 0.948, *d* = 0.21, 95% CI [−0.04, 0.47], $BF_{10}$ = 0.057. Directly comparing locations revealed a more negative criterion for the high- compared to the low-value target, *t*(59) = 1.78, *p* = 0.04, *d* = 0.23, 95% CI [0.03, 0.49], $BF_{10}$ = 1.19 (anecdotal evidence), indicating a bias in line with the value

manipulation. However, any effect in the criterion may reflect either a response bias or a perceptual bias.

**Motivation quantity biases explicit but not implicit perceptual measures (Experiment 2)**
Experiment 1 could not distinguish whether motivation induces a perceptual bias or a response bias. Experiment 2 aimed to disentangle these two possibilities using an implicit behavioral readout. We simultaneously assessed perceptual reports and pursuit eye movements as an implicit measure of motion perception, allowing us to compare biases obtained from perceptual reports with biases observed in pursuit. Although pursuit and motion perception can be dissociated[57–60], they typically show a substantial agreement[61–63]: First, pursuit follows the perceived motion rather than the retinal stimulus[64]. Second, compared to perception, pursuit shows a similar[52] or even superior[65] sensitivity to velocity changes. Third, both can be successfully modeled as resulting from the combination of sensory input with a zero-velocity prior[66–68], suggesting that both operate at a comparable level in the visual hierarchy. Hence, we reasoned that a perceptual bias should manifest in both perceptual reports and in pursuit, whereas a response bias should affect reports only.

Participants pursued two subsequently moving dots with their eyes and indicated which moved faster (Fig. 4A, B). In an unbiased block, participants performed the task without any manipulation regarding motivation. In this unbiased condition, smooth pursuit and motion perception showed a

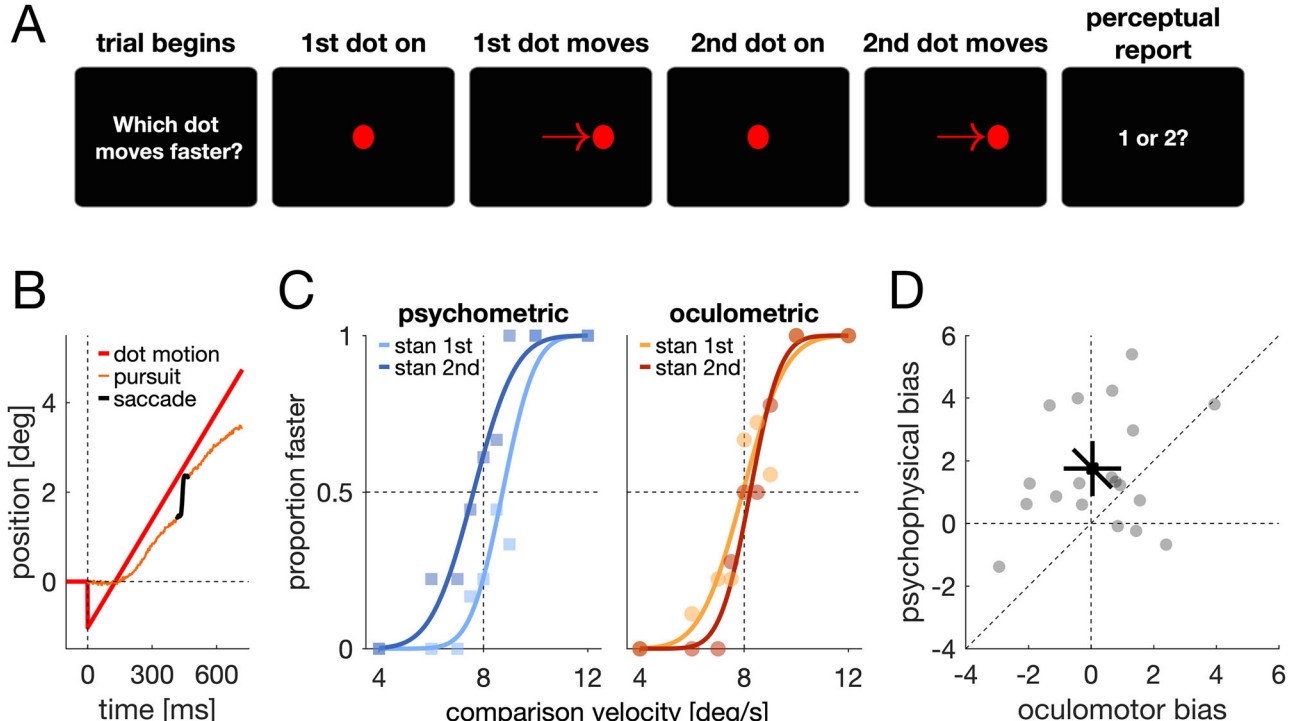

**Fig. 4 | Experiment 2: motivation biases perceptual reports but not implicit measures of perception. A** Trial procedure. Each trial requires participants to pursue two moving dots and decide which of the two moved faster. **B** Pursuit trace (orange) intervened by a catch-up saccade (black) relative to the moving dot (red). **C** Psychometric (blue, left panel) and oculometric (orange, right panel) functions of one participant in the biased block. This example participant received a higher reward when correctly indicating that the first dot moved faster. A separate function was fit for each stimulus order (standard stimulus first, lighter colors, or standard stimulus second, darker colors). The psychophysical (oculomotor) bias is derived from the difference in PSE values between the two psychometric (oculometric) functions relative to the same difference from the unbiased block. **D** Psychophysical and oculomotor bias. The black data point denotes the overall mean with 95% confidence intervals. The diagonal error bar marks the within-participant variability between the two bias indices and must be compared to the identity line. Gray circles denote data from individual participants ($n = 20$ participants).

substantial agreement, $r(18) = 0.95$, $p < 0.001$, $BF_{10} = 211$. In a second block, we manipulated participants' motivation by rewarding them for a correct response on perceived dot speed, and each participant was motivated to either perceive the first or the second dot as moving faster.

We computed bias indices from perceptual reports (psychophysical bias) and from pursuit (oculomotor bias) (Fig. 4C). The psychophysical bias ($M_{psych} = 1.75$, $SD_{psych} = 1.88$), was different from zero, $t(19) = 4.165$, $p < 0.001$, $d = 0.93$, 95% CI [0.40, 1.45], $BF_{10} = 64.1$ (Fig. 4D), highlighting the presence of a bias. The presence of a similar bias in the oculomotor domain would be indicative of a perceptual bias. Conversely, the absence of a bias in the oculomotor domain would instead indicate a response bias. Crucially, we observed no bias in oculomotor responses, suggesting that the bias found in perceptual reports can be attributed to a response bias. Specifically, the mean oculomotor bias, ($M_{oculo} = 0.04$, $SD_{oculo} = 1.96$), was significantly smaller than the psychophysical bias, $t(19) = 2.897$, $p = 0.009$, $d = 0.65$, 95% CI [0.16, 1.13], $BF_{10} = 5.476$, and indistinguishable from zero, $t(19) = 0.10$, $p = 0.921$, $d = 0.02$, 95% CI [−0.42, 0.46], $BF_{10} = 0.233$, consistent with motivation biasing response behavior but inconsistent with motivation biasing perception. Thus, we conclude that the observed psychophysical bias reflects a response bias, not a perceptual bias.

## Gaze behavior determines perception in ambiguous stimuli (Experiments 3 and 4)

The first two experiments showed effects of motivation on perceptual reports but did not provide any evidence that motivation biases perception or changes perceptual sensitivity. This contrasts with studies presenting evidence for motivated perception[6,20]. We reasoned that a possible explanation for this discrepancy might be the combination of ambiguous stimuli with differences in gaze position. Based on our finding that motivation changes gaze position (Experiment 1; Fig. 3), we hypothesized that when using ambiguous stimuli, for example, drawn visual illusions[6] or composite images[20,21,51] (i.e., morphed face-house images), people may select different gaze positions based on their behavioral goal. These different gaze positions may, in turn, produce different percepts. Under this view, motivation does not bias how the incoming sensory input is interpreted; it rather changes the sensory input itself by behavioral selection.

We tested this hypothesis using three complementary approaches. Our first approach examined whether spontaneous perceptual reversals are preceded by changes in gaze position. In our second approach, we tested whether participants select different gaze positions when intentionally biasing their perception. Our third approach assessed whether perceptual biases emerge when gaze position is experimentally enforced. The first two approaches were tested in Experiment 3, and the third in Experiment 4.

In Experiment 3 (Approach 1), participants freely viewed composite face/house images or ambiguous visual illusions while continuously reporting their perception. In composite images, perceptual reversals were preceded by a shift in gaze position (Fig. 5A), $t_{sum} = 7300$, $t_{crit} = 765.8$, $p < 0.001$, time window: [−328 ms to 1000 ms]. Specifically, participants reported perceiving the face as more dominant, shortly after looking at the eye region of the face. Conversely, perceiving the house as dominant was preceded by gaze shifts away from the eye region. Similar patterns were observed with all illusions (Fig. 5B), seal-donkey, $t_{sum} = 6366$, $t_{crit} = 799.5$, $p < 0.001$, time window: [−282 ms to 961 ms], duck-rabbit, $t_{sum} = 5119$, $t_{crit} = 805.5$, $p < 0.001$, time window: [−272 ms to 1000 ms], young-woman-old-woman, $t_{sum} = 4253$, $t_{crit} = 647.8$, $p < 0.001$, time window: [−355 ms to 682 ms], and B-13, $t_{sum} = 1086$, $t_{crit} = 581.2$, $p = 0.014$, time window: [−67 ms to 196 ms]. This suggests that systematic differences in gaze position are preceding differences in perception.

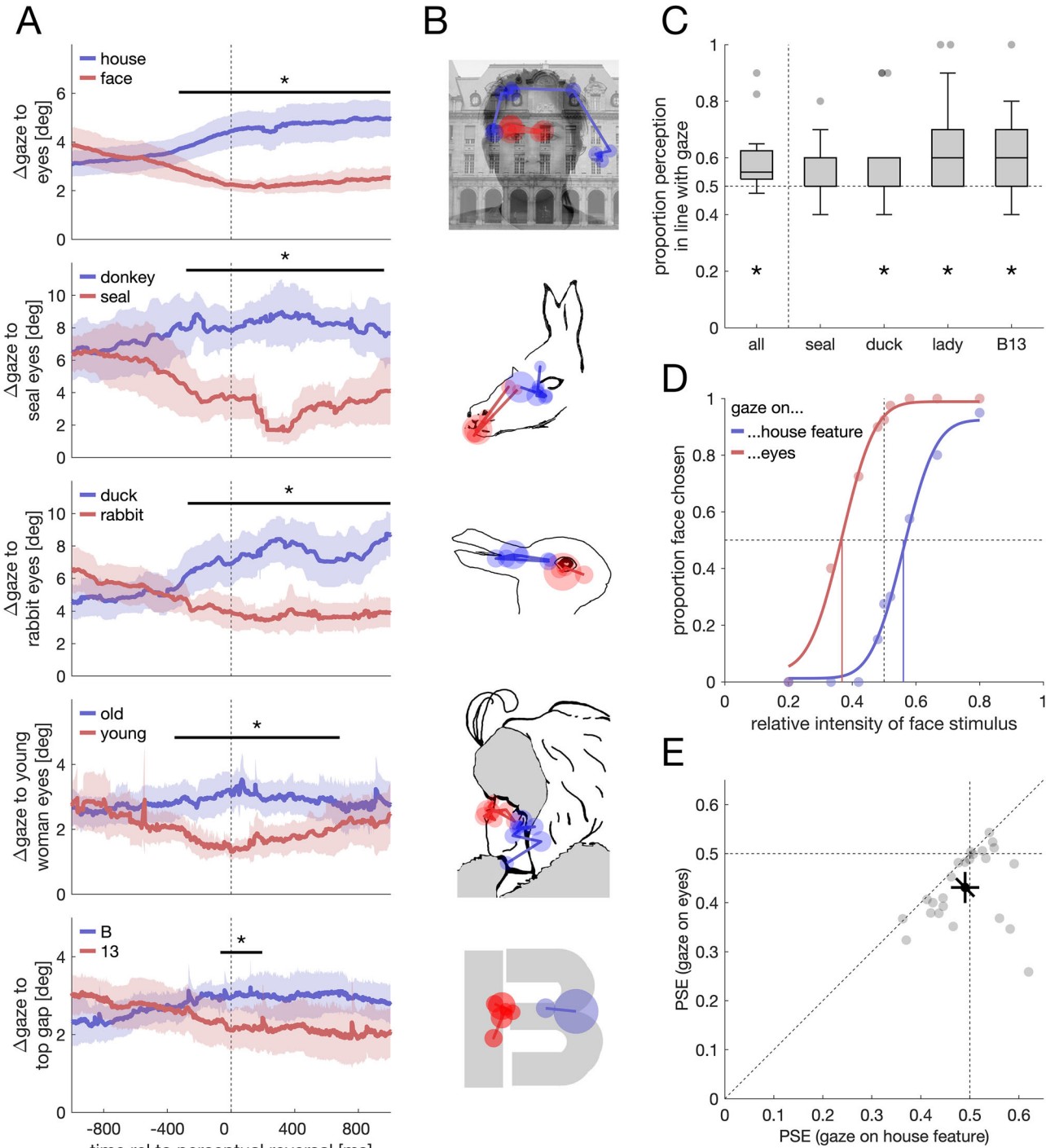

**Fig. 5 | Experiments 3 and 4: Gaze behavior explains perception in ambiguous stimuli. A** Changes in gaze position precede spontaneous perceptual reversals. Colors denote the dominant percept after the reversal (time: 0 ms). Shaded areas are 95% confidence intervals. Lines and asterisks indicate a significant difference as revealed by a cluster-permutation test (*n* = 16 participants). **B** Stimuli and fixation maps from representative individual participants when asked to bias their perception to one of two possible interpretations (red: face, seal, rabbit, young woman, 13; blue: house, donkey, duck, old woman). **B** Each map depicts fixations from a single participant (participants differ across panels). Each disc represents one fixation, with the disc size scaling with fixation duration. Composite stimuli (top panel) were created by merging face and house images. Face images were taken from the Chicago Face Database (CFD)[54], used with permission. House images were taken from the SUN Database[55]. The illusion stimuli depicted in the figure were redrawn by the authors and are not the actual illusion stimuli used in the experiments. **C** Boxplots showing the proportion perceptual reports were in line with the forced-fixation manipulation in the visual illusion stimuli (*n* = 24 participants). Asterisks denote a significant difference from chance (*t*-test and Wilcoxon signed-rank). **D** The effect of forced fixations on composite stimuli. Example psychometric functions from one participant. A separate psychometric function was fit when participants were forced to fixate on the eye region of the face (red) or on a salient house feature (blue). Colored vertical lines indicate the points of subjective equality. **E** Scatter plot of PSE values (*n* = 24 participants). The black data point denotes the overall mean with 95% confidence intervals. The diagonal error bar marks the within-participant variability and must be compared to the identity line. Gray circles denote data from individual participants.

Within Experiment 3 (Approach 2), we also tested whether participants can voluntarily bias their perception of the same ambiguous stimuli by selecting different gaze positions. We asked participants to voluntarily bias their perception to one of the two alternatives. Importantly, eye movements or gaze position were not mentioned in the instructions. We tracked participants' gaze (Fig. 5B) and compared gaze positions when asked to perceive one percept (e.g., the seal) or the other (e.g., the donkey). Gaze positions differed for composite images, $\Delta gaze = 2.85°$, $\Delta gaze_{crit} = 0.17°$, $p < 0.001$, the seal-donkey illusion, $\Delta gaze = 6.21°$, $\Delta gaze_{crit} = 2.75°$, $p < 0.001$, the duck-rabbit illusion, $\Delta gaze = 4.63°$, $\Delta gaze_{crit} = 3.29°$, $p < 0.001$, the young-woman-old-woman illusion, $\Delta gaze = 2.50°$, $\Delta gaze_{crit} = 1.93°$, $p < 0.001$, and the B-13 illusion, $\Delta gaze = 2.37°$, $\Delta gaze_{crit} = 1.69°$, $p < 0.001$. This suggests that participants can voluntarily bias their perception by selecting a different gaze position.

Experiment 4 (Approach 3) then tested the effect in the reverse direction by examining whether perception is biased when enforcing fixation on diagnostic image regions. Across the four visual illusions presented, perception was biased in line with the gaze manipulation (Fig. 5C), $t(23) = 4.12$, $p < 0.001$, $d = 0.84$, 95% CI [0.37, 1.30], $BF_{10} = 76.45$. Significant differences were found for B-13, $t(23) = 4.16$, $p < 0.001$, $d = 0.85$, 95% CI [0.37, 1.31], $BF_{10} = 83.41$, young-woman-old-woman, $t(23) = 3.73$, $p = 0.001$, $d = 0.76$, 95% CI [0.30, 1.21], $BF_{10} = 32.27$, duck-rabbit, $t(23) = 2.22$, $p = 0.036$, $d = 0.45$, 95% CI [0.03, 0.87], $BF_{10} = 1.68$ (anecdotal evidence), but not for the seal-donkey figure, $t(23) = 1.68$, $p = 0.107$, $d = 0.34$, 95% CI [−0.07, 0.75], $BF_{10} = 0.72$ (anecdotal evidence).

Lastly, we tested the same notion for composite stimuli. When participants were forced to fixate in-between the eyes, a lower face signal was sufficient for both categories to be perceived as equally strong ($M_{eyes} = 0.43$, $SD_{eyes} = 0.075$) compared to a fixation on a salient house feature, ($M_{house} = 0.49$, $SD_{house} = 0.068$), $t(23) = 3.28$, $p = 0.003$, $d = 0.67$, 95% CI [0.22, 1.11], $BF_{10} = 12.51$ (Fig. 5D, E). This shows that differences in gaze position do not only covary with changes in perception, but that differences in gaze position are causal in biasing perception, both for visual illusions and composite stimuli. Thus, all three approaches provided converging evidence that perceptual biases in ambiguous stimuli can be explained by differences in gaze position.

### No evidence for an influence of motivational quality

Across all four experiments, motivational quality was unrelated to all perceptual and behavioral measures. Further, motivational quality did not modulate the effect of motivational quantity. Hence, we did not find evidence in support of the idea that motivational quality contributes to the effects observed in the present studies. The corresponding results can be found in the Supplementary Figs. 1–4.

## Discussion

Our results provided no evidence for a direct link between motivation and perception. Instead, motivation induced a response bias (Fig. 4) and biased gaze behavior, which in turn modified perceptual sensitivity (Fig. 3) and explained perceptual biases in ambiguous stimuli (Fig. 5). Thus, our findings contradict the motivated perception hypothesis[3,6,20] and provide a clear alternative explanation for many of the findings in this literature: Rather than changing how the brain interprets the incoming sensory information, motivation changes how the world is visually sampled and thus alters the sensory input itself.

The idea that our desires, goals, and motivation change how we perceive the world can be traced back to the New Look movement during the middle of the last century[69]. However, these early claims were disproven and explained by methodological shortcomings[70]. Ever since these ideas reappeared around the beginning of the current century[5,6], there has been an intense debate as to whether we really do see what we wish to see or whether these findings reflect response biases[12–14,16,71]. Our study builds upon this work, addressing motivated perception from a cognitive neuroscience perspective[19,20,29]. By integrating earlier findings on the interplay between motivation and gaze[24–26,72], and between gaze and perception[73–76], we provide

evidence that many phenomena that have been attributed to top-down modulation of perception by motivation may not be as surprising as they initially seem. Instead, these findings can be explained by response biases and differences in gaze position. Importantly, we show for the same set of stimuli that have been used in motivated perception studies that (i) perception is biased depending on the gaze position, that (ii) people select different gaze positions when they intentionally control their perception, and that (iii) perceptual reversals are preceded by changes in gaze position – an observation that has also been made for binocular rivalry[74,77]. Interestingly, response and gaze behavior can also alter activity in the brain areas previously reported to be associated with motivated perception[20,27,78–83]. Our results, therefore, raise the question as to whether neuroimaging evidence for motivated perception may similarly be explained by these mechanisms.

The simultaneous recording of gaze position and spatial attention as possible mediators between motivation and perception allowed us to differentiate between potential explanatory mechanisms underpinning purported motivated perceptual effects. While gaze position was the main factor driving perceptual sensitivity (Fig. 5F), our measure of spatial attention was unrelated to reports and perception. We aimed to obtain an attentional measure independent of eye movement preparation by presenting attentional probes before the onset of any peripheral stimuli that could prompt a gaze shift. This allowed us to detect whether one target location was attentionally depreciated throughout the experiment. The results suggested this was not the case. However, this does not rule out the involvement of attentional mechanisms; rather, it suggests that attention shifts may have occurred after stimulus onset, in which case they would likely be reflected in eye movement endpoints[84,85].

Our study concurrently assessed two facets of motivation: we measured the self-concordant quality of each participant's motivation and manipulated the quantity of motivation for perceiving alternative outcomes by changing the reward structure of the task. Unlike motivational quantity, motivational quality had no influence on perception and behavior (Supplementary Figs. 1–4). One explanation for these results is that motivational value immediately influences behavior and is therefore reflected in short experimental tasks, whereas motivational quality is more important for sustained goal regulation[41]. While the selected tasks served to study motivated perception with a high degree of experimental control, the tasks themselves were difficult to reconcile with complex, real-world goals (e.g., having a successful career) – the typical target domain of the used self-report instrument[43].

### Limitations

Since we had no experimental control over the questionnaire data, one weakness of our approach to measure motivational quality was that our sample was quite homogeneous and reported relatively high self-concordant motivation (115/120 participants had positive self-concordance indices, reflecting internalized motivation). It is possible that wider variability in motivation quality would have produced more pronounced differences; however, given that the results across multiple samples consistently suggest that the degree of one's self-concordant motivation for a task does not contribute to motivated perception in the present experiments, we consider it unlikely that the results would have changed with a more balanced sample. Despite these limitations, we contend that our consideration of both motivational quality and quantity represents a more encompassing view of motivation than has been taken in the past – a view that is in line with the contemporary conceptualization of motivation in the social sciences[38,39,86].

### Perception is shaped by goal-directed behavior

This leaves us with the question of what is perception? In our view, perception is the process of constructing an understanding of the world based on the activity of our sensors. However, this sensor activity is shaped by an inherently active, goal-directed process: We move our hands so that our skin receptors provide most information about the material properties we want to estimate[87], we turn our head to better localize sound sources[88], and we

move our eyes to that detail of the visual world that is relevant for us or that we consider interesting[34,35,89]. Our research suggests that differences in perception do not come about because the perceptual systems of different people interpret the same sensory input differently, colored by desire or motives. On the contrary, it is this second process, re-directing our sensors in a goal-directed manner, that can explain individual differences in perceptual judgments when people are confronted with the same situation. A radiologist might move her eyes differently than her colleague, depending on her search strategy or depending on what she expects to find[90,91], causing her to perceive what her colleague misses. An experienced driver might monitor the road and traffic differently[92], being more sensitive to dangerous situations. And an expert baseball player might know how to control his gaze to gather all the relevant visual information to hit the ball[93]. All these scientifically examined real-world scenarios describe cases in which people control their gaze in a top-down manner, thereby altering what they perceive. Our work adds to this by demonstrating that although gaze patterns are influenced by goals and motivation, the perceptual construction process itself is not.

## Conclusion

In conclusion, when you watch soccer with others and shout "Goal!" while someone rooting for the other team insists that the ball did not cross the line, the disagreement may not stem from fundamentally different perceptions of the same sensory input. Rather, you might have been looking at different aspects of the event: perhaps your eyes were drawn to the goal line, while the other person was focused on the ball. Even if you both happened to focus on the same part of the scene and perceive the event in the same way, your motivation – hoping for a goal – may have biased your response to what you saw. Our findings suggest that such disagreements are not necessarily due to perceptual distortions – the ball does not appear larger[8]; the goal line does not appear closer[9] – but rather to differences in overt attention and in how people respond to ambiguous situations, thereby identifying the behavioral pathway through which motivational states influence perceptual judgments. In short, motivation doesn't change how our brains make sense of the sensory input – it changes where we look, and what we say.

## Data availability

The data of all experiments (primary data as well as aggregated data) are publicly available. Data and material (including the German version of the C-RAI questionnaire) can be accessed via the Open Science Framework (https://doi.org/10.17605/OSF.IO/2S5V6).

## Code availability

Analysis scripts reproducing the Figures are publicly available. The code was written and tested using MATLAB R2021a. Scripts can be accessed via the Open Science Framework (https://doi.org/10.17605/OSF.IO/2S5V6).

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

## Acknowledgements

This work was funded by the DFG (German Research Foundation), Project No. 427754309, awarded to C.W. Scientific exchange was funded by the DAAD (German Academic Exchange Service), Project No. DAAD57654886, awarded to H.R. and M.L. The funders had no role in study design, data collection and analysis, decision to publish, or preparation of the manuscript. The authors thank Linda Tackenberg for collecting the data.

## Author contributions

C.W., M.L., and H.R. conceptualized the study. C.W. and H.R. analyzed the data. C.W. visualized the data and wrote the first manuscript draft. C.W., M.L., and H.R. revised the manuscript and approved the final manuscript version for submission.

## Funding

## Competing interests

The authors declare no competing interests.
