## [Transparent Peer Review file · Communications Psychology]

Motivation biases behavior but not perception

Corresponding Author: Dr Christian Wolf

Version 0:

Decision Letter:

Dear Dr Wolf,

Thank you for your patience during the peer-review process. Your manuscript titled "Motivation biases behavior – not perception" has now been seen by 3 reviewers, and I include their comments at the end of this message. They find your work of interest but raised some important points. We are interested in the possibility of publishing your study in Communications Psychology, but would like to consider your responses to these concerns and assess a revised manuscript before we make a final decision on publication.

We therefore invite you to revise and resubmit your manuscript, along with a point-by-point response to the reviewers. Please highlight all changes in the manuscript text file.

Editorially, though we acknowledge the general enthusiasm of reviewers about your study, we consider that it is important that you address all requests for clarification and readability improvement.

I am attaching an Editorial Requests Table that details critical reporting requirements for the revised manuscript. Please attend to each item and ensure your manuscript is fully compliant. If your revised manuscript is not aligned with these requests on major issues, such as those concerning statistics, it may be returned to you for further revisions without re-review.

Please submit the following items:

- Revised manuscript
- Point-by-point response to the referees' comments
- Cover letter (as a separate document)
- <https://www.nature.com/documents/nr-reporting-summary.pdf> Nature Research Reporting Summary
- Completed Editorial Request Table (attached).

via this link: Link Redacted .

Additional guidance is available in our style and formatting guide Communications Psychology formatting guide.

Best regards,

Troby Lui, on behalf of

Mael Lebreton

Troby Lui, PhD
Associate Editor
Communications Psychology

Mael Lebreton, PhD
Editorial Board Member
Communications Psychology
orcid.org/0000-0002-2071-4890

REVIEWER EXPERTISE:

Reviewer #1: (motivated) perception
Reviewer #2: (motivated) perception; Motivated beliefs
Reviewer #3: Motivated beliefs

REVIEWER REPORTS:

Reviewer #1 (Remarks to the Author):

Review of manuscript COMMSPSYCHOL-25-0628 titled "Motivation biases behavior – not perception" for Communications Psychology.

In this manuscript the authors examine the question of whether motivation can truly bias perception, or whether these biases act at the level of behaviour, via active sampling of the perceptual stimulus (for example, by choosing where to look). The authors employ a rigorous and creative set of experiments with sophisticated analyses, showing convincing evidence that in fact motivational biases in perception are not truly perceptual but mediated by behaviour. In experiment 1 they show that motivational biases in perceptual sensitivity can be explained by biases in eye movements. In experiment 2 they use a novel approach of comparing smooth pursuit eye-movements as an implicit measure of perceived motion speed to dissociate response from perceptual bias. In Experiments 3 and 4 they show how fixation location determines the perception of ambiguous stimuli and discrimination of composite (face/house) images. Each of these experiments provides convincing support for motivational biases in behaviour and explains previous findings suggestive of motivational biases in perception. These findings will be very interesting to a wide audience across psychology, and especially cognitive and perceptual disciplines. The science presented in this manuscript is excellent, I have no suggestions for improvement.

My only issue is of access and read-ability, which I think can be easily improved. The methods are very dense, and often confusing. After reading the methods it is difficult to remember experiment 1 for the results, although the reminder explanations in the results are very helpful. I get the impression the paper was originally written with the methods at the end. For example, at the beginning of the methods the authors write that their sample size was determined by a power analysis (line 137) "to distinguish between a serial and a parallel mediation model (see Supplementary Material)." But the notion of serial and parallel has not been mentioned in the introduction. Actually it is not mentioned in the rest of the manuscript, perhaps the whole sample size justification could be moved to the supplementary if it is now not relevant for the current results?

In this format, it would be helpful to have a figure of the methods for all experiments together. This could also help reduce the length, as the descriptions of the ambiguous stimuli could be left to the figure legend. This should also help the reader grasp the multiple complex experiments, for example, for experiments 3 and 4 have an intermixed description. For experiment 3 it is written (line 350) "Each trial in turn consisted of three different phases. Preceding the first phase..." then this pre-phase

and phase one are described. Then for phase two and three it is written "In phases two and three ("approach two")" and then for Experiment 4 it is written "Experiment 4 ("approach three)". So Experiment 3 has a pre-phase and then 3 phases with two approaches and experiment 4 has just the third approach. It's difficult to keep track of these phases and approaches.

I think it could be most useful for the reader if the work could be presented as: Experiment 1 methods, results; Experiment 2 methods, results; Experiments 3 and 4 methods, results. But I'm unsure if the journal allows this.

The other concern is the overall length. The journal suggests 5000 words, I think this is about twice that. I think everything written is important, there is limited room for reducing length by being more concise (although the introduction could benefit from some pruning). Perhaps more can be moved to the supplementary?

My only other concern for read-ability/access is the example at the beginning and the end of the manuscript is both long (not helping with the length of the manuscript) but also contains a lot of technical jargon outside of the expected expertise of the reader. How does my favourite team "attack" and what does it mean to "score a goal"? What makes the other player "defensive"? What is a "foul"? What is a "Dive"?

Reviewer #2 (Remarks to the Author):

This manuscript addresses an enduring and significant debate in cognitive science regarding whether and how motivation influences perception using a series of clever experiments that combine psychophysics with eye-tracking.

In study 1, participants were asked to detect whether a numerical digit was present in a noisy stimulus across two locations, with one location linked to a higher reward for correct detection. Results showed that participants exhibited a stronger bias to report the digit as present at the high-value location compared to the low-value location, but any biases in perceptual sensitivity were explained by differences in eye-movements.

In study 2, participants were required to indicate which of two moving dots appeared faster, with a higher reward for correct responses associated with one of the dots. Here, reward magnitude was found to bias perceptual reports but not smooth pursuit eye-movements.

In studies 3 and 4, participants were presented with ambiguous visual images. Systematic differences in eye-gaze preceded perceptual shifts. When participants were asked to perceive a particular percept, their gaze was directed to the corresponding features of the stimulus. Furthermore, when instructed to direct their gaze to those features, participants were biased to report the corresponding percept.

For context, I had previously reviewed this manuscript at a different outlet. I found this version of the manuscript significantly improved. I particularly appreciated the addition of Studies 3 and 4, as they provide a convincing demonstration of how motivational effects on perceptual judgments can be mediated by differences in eye-movements. Pending some minor revisions for clarification, I would be happy to recommend publication.

1. My main comment is that manuscript isn't always clear on the criteria of what counts as motivational influences on perception. I thought the paragraph in the discussion about "what is perception?" is a valuable addition, but this more nuanced perspective seems at odds with introduction that adopts a stricter criterion.

My recommendation would be to explicitly distinguish between "direct" vs. "mediated" effects of motivation early in the manuscript, which I believe would also enhance the contribution of the current results, where the authors show *how* motivational effects on eye-movements can explain differences in perceptual judgments.

More generally, I think the field would benefit greatly from moving past the narrow question of whether motivation biases perception (the answer to which will likely depend on what one means by "motivation", "bias", and "perception", as well as the experimental context), and toward understanding the boundary conditions, pathways, and concrete perceptual differences, which I think this paper does so excellently.

2. On a related note: Line 99: "Additionally, motivation might not necessarily bias perception, but might instead modulate perceptual sensitivity – that is, how well something is perceived rather than what is perceived."

Wouldn't a modulation of perceptual sensitivity also be a form of bias in perception? Seeing something more clearly seems like it would count as perceptual effect.

3. Lines 419: "We used signal-detection theory to analyze the response behavior of every individual (Fig. 2A-B) and to distinguish between bias (criterion) and sensitivity (d'). Mean d' values were $M_{high} = 0.63$ ($SD_{high} = 0.56$) and $M_{low} = 0.45$ ($SD_{low} = 0.49$) and were not larger for the high compared to the low value location, $t(59) = 1.68$, $p = 0.050$, $d = 0.22$, $BF_{10} = 0.997$ (Fig. 3B). Thus, strictly speaking, we found no difference in perceptual sensitivity between the two locations. However, without controlling for any mediating mechanisms, a tendency for better sensitivity at the high value location is apparent."

I thought the last two sentences were written in a somewhat confusing manner, as it's not immediately clear what is meant by "strictly speaking" and "tendency". Perhaps it would be helpful to be more explicit in saying that the evidence is close but did

not cross predetermined statistical threshold, and that authors will later show that any apparent motivational effect is entirely explained by eye movements?

Minor:

Figure 5B: Could the authors clarify if the fixation maps are from a single participant or the whole group?

Line 524: typo in caption of panel C: Psychometric [and] oculometric.

Reviewer #3 (Remarks to the Author):

This paper seeks to elucidate the source of "motivated-perception" effects by carefully considering both motivation and perception. Experiments 3 and 4, in particular provide useful evidence concerning a role of gaze behavior in effects that might otherwise be hard to interpret.

In Experiment 1, the main finding is that recognizing numbers in noise in two locations shows clear evidence that sensitivity is affected by gaze direction, which leads to biased performance. In Experiment 2 further evidence is adduced for separation of gaze behavior and perception. But it is really in Experiments 3 and 4 that the authors look at how gaze corresponds with observers reports of perception and also with their strategies to obtain certain perceptual information.

This is a very clear paper, which makes a clear argument. It is possible to interpret the paper as providing a means by which responses may be biased by attentional selection (gaze is overt attention), which is, of course, not surprising, but nice to see so clearly demonstrated. The stronger argument of the paper is that many "motivational" biasing effects on perception are fairly cognitive (e.g., based on attentional differences, rather than perceptual differences.) This seems like an important addition to the literature.

Signed review,
Frank Durgin

Communications Psychology is committed to improving transparency in authorship. As part of our efforts in this direction, we are now requesting that all authors identified as 'corresponding author' create and link their Open Researcher and Contributor Identifier (ORCID) with their account on the Manuscript Tracking System prior to acceptance. ORCID helps the scientific community achieve unambiguous attribution of all scholarly contributions. You can create and link your ORCID from the home page of the Manuscript Tracking System by clicking on 'Modify my Springer Nature account' and following the instructions in the link below. Please also inform all co-authors that they can add their ORCIDs to their accounts and that they must do so prior to acceptance.
<https://www.springernature.com/gp/researchers/orcid/orcid-for-nature-research>

Version 1:

Decision Letter:

Dear Dr Wolf,

Your manuscript titled "Motivation biases behavior – not perception" has now been seen by our reviewers, whose comments appear below. In light of their advice I am delighted to say that we are happy, in principle, to publish a suitably revised version in Communications Psychology.

We therefore invite you to revise your paper one last time to address the remaining concerns of our reviewers and a list of editorial requests. At the same time we ask that you edit your manuscript to comply with our format requirements and to maximise the accessibility and therefore the impact of your work.

EDITORIAL REQUESTS:

SUBMISSION INFORMATION:

OPEN ACCESS:

* DATA AVAILABILITY:

Link Redacted

Best regards,

Troy Lui

Troy Lui, PhD
Associate Editor
Communications Psychology

Mael Lebreton, PhD
Editorial Board Member
Communications Psychology
orcid.org/0000-0002-2071-4890

REVIEWERS' COMMENTS:

Reviewer #1 (Remarks to the Author):

The authors have fully addressed my previous comments. I congratulate them on an excellent manuscript.

Reviewer #2 (Remarks to the Author):

The authors have fully addressed my comments. This is a valuable addition to the literature, and I am happy to recommend acceptance.

Reviewer #3 (Remarks to the Author):

I had no requests regarding the previous version. It seems that the constructive suggestions of the other reviewers have been addressed.

We thank the editors and the reviewers for taking the time to assess our manuscript and for the valuable feedback provided. Below is a point-by-point response to the issues raised by the reviewers. Our responses are indented and written in italics. Quotes from the revised manuscript are written in red, with line numbers referring to the manuscript with tracked changes. Changes to the manuscript are additionally highlighted using an underscore.

Christian Wolf, Markus Lappe, Hugh Riddell

Reviewer #1 (Remarks to the Author):

Review of manuscript COMMSPSYCHOL-25-0628 titled “Motivation biases behavior – not perception” for Communications Psychology.

In this manuscript the authors examine the question of whether motivation can truly bias perception, or whether these biases act at the level of behaviour, via active sampling of the perceptual stimulus (for example, by choosing where to look). The authors employ a rigorous and creative set of experiments with sophisticated analyses, showing convincing evidence that in fact motivational biases in perception are not truly perceptual but mediated by behaviour. In experiment 1 they show that motivational biases in perceptual sensitivity can be explained by biases in eye movements. In experiment 2 they use a novel approach of comparing smooth pursuit eye-movements as an implicit measure of perceived motion speed to dissociate response from perceptual bias. In Experiments 3 and 4 they show how fixation location determines the perception of ambiguous stimuli and discrimination of composite (face/house) images. Each of these experiments provides convincing support for motivational biases in behaviour and explains previous findings suggestive of motivational biases in perception. These findings will be very interesting to a wide audience across psychology, and especially cognitive and perceptual disciplines. The science presented in this manuscript is excellent, I have no suggestions for improvement.

We thank Reviewer 1 for the detailed and constructive feedback. The reviewer’s comments were instrumental in improving the readability and structure of the manuscript. In particular, they motivated us to substantially shorten and streamline several sections, resulting in a more focused and accessible presentation of our results.

My only issue is of access and read-ability, which I think can be easily improved. The methods are very dense, and often confusing. After reading the methods it is difficult to remember experiment 1 for the results, although the reminder explanations in the results are very helpful. I get the impression the paper was originally written with the methods at the end. For example, at the beginning of the methods the authors write that their sample size was determined by a power analysis (line 137) “to distinguish between a serial and a parallel mediation model (see Supplementary Material).” But the notion of serial and parallel has not been mentioned in the introduction. Actually it is not mentioned in the rest of the manuscript, perhaps the whole sample size justification could be moved to the supplementary if it is now not relevant for the current results?

We thank the reviewer for this helpful suggestion. We agree that the sample size justification referring to serial versus parallel mediation models is no longer central

to the current manuscript and may impair readability. We therefore moved the detailed power analysis to the Supplementary Material and streamlined the Participants section to focus on information essential for reproducibility. We hope this change improves the clarity and accessibility of the Methods section.

In this format, it would be helpful to have a figure of the methods for all experiments together. This could also help reduce the length, as the descriptions of the ambiguous stimuli could be left to the figure legend [...].

We thank the reviewer for this thoughtful suggestion. We agree that a joint methods figure can be very helpful for orienting readers in complex multi-experiment studies, and we carefully considered this option during the revision.

After weighing the advantages and disadvantages, we decided not to introduce a separate, experiment-spanning methods figure. As the reviewer points out, our methods span several distinct methodologies and it was our view that presenting these procedures together created more confusion than clarity. As mentioned, we have made changes to the methods sections to improve their interpretability and present figures demonstrating the related experimental procedure with each relevant methods section.

Inspired by the comment of the reviewer, we streamlined the stimulus descriptions of Experiment 1 in the Methods by moving purely illustrative details (e.g., example stimulus variants) to the corresponding figure captions (Figure 1), while keeping all parameters required for reproducibility in the main text, in line with the journal's guidelines.

For Experiments 2–4, the associated figures (Figure 4 and Figure 5) intentionally integrate a schematic reminder of the experimental setup and/or the stimuli with the corresponding results. The current figures therefore support memorability by providing methodological context exactly at the point where readers interpret the data, rather than requiring them to refer back to a separate overview figure. We were concerned that decoupling methods and results would make it harder – rather than easier – for readers to follow the logic of each experiment. We therefore decided to leave the figures as they are.

[...] This should also help the reader grasp the multiple complex experiments, for example, for experiments 3 and 4 have an intermixed description. For experiment 3 it is written (line 350) “Each trial in turn consisted of three different phases. Preceding the first phase...” then this pre-phase and phase one are described. Then for phase two and three it is written “In phases two and three (“approach two”)” and then for Experiment 4 it is written “Experiment 4 (“approach three”)”. So Experiment 3 has a pre-phase and then 3 phases with two approaches and experiment 4 has just the third approach. It’s difficult to keep track of these phases and approaches.

We agree that the previous description mixed phases, approaches, and experiments in a way that was difficult to follow. We therefore revised the Procedure section of Experiment 3 to provide a clear description of the experiment's hierarchical structure (each trial comprises three phases), without explicitly mentioning the different approaches. Additionally, we revised the results section focusing on the three different approaches, and clarifying how Experiments 3 and 4 jointly test the same hypothesis.

Lines 1007 – 1015:

***Procedure and design.** Experiment 3 consisted of 34 trials, each comprising three phases. At the beginning of each trial, text indicated the two possible percepts (e.g.: “seal versus donkey”). In Phase 1 (spontaneous perception), participants freely viewed the stimulus and continuously reported the currently dominant percept via button press. This phase captured spontaneous perceptual reversals and ended when participants pressed the space bar. In Phases 2 and 3 (intentional bias) participants were instructed to bias their perception toward one of the two possible percepts. Once successful, participants pressed the space bar and were asked to maintain their percept for the remaining 2500 ms of stimulus presentation. Trial order was randomized across participants.*

Lines 1551 – 1557:

We tested this hypothesis using three complementary approaches. Our first approach examined whether spontaneous perceptual reversals are preceded by changes in gaze position. In our second approach, we tested whether participants select different gaze positions when intentionally biasing their perception. Our third approach assessed whether perceptual biases emerge when gaze position is experimentally enforced. The first two approaches were tested in Experiment 3, and the third in Experiment 4. In Experiment 3 (Approach 1), participants...

Lines 1633 – 1636:

Within Experiment 3 (Approach 2), we also tested whether participants can voluntarily bias their perception of the same ambiguous stimuli by selecting different gaze positions. We asked participants to voluntarily bias their perception to one of the two alternatives. Importantly, eye movements or gaze position were not mentioned in the instruction. We tracked participants’ gaze...

Line 1644:

Experiment 4 (Approach 3) then tested the effect in the reverse direction by examining whether perception is biased when enforcing fixation on diagnostic image regions.

I think it could be most useful for the reader if the work could be presented as: Experiment 1 methods, results; Experiment 2 methods, results; Experiments 3 and 4 methods, results. But I’m unsure if the journal allows this.

We thank the reviewer for this helpful suggestion. After consulting with the editorial team, we were asked to adhere to the journal’s required Introduction–Methods–Results–Discussion format.

To improve readability within this format, we have revised the manuscript with a focus on improving clarity and reader guidance, including clearer structuring of the Methods and Results sections and more explicit reminders of the experimental designs where appropriate. We hope that these changes sufficiently address the reviewer’s concerns regarding readability and memorability.

The other concern is the overall length. The journal suggests 5000 words, I think this is about twice that. I think everything written is important, there is limited room for reducing length by being more concise (although the introduction could benefit from some pruning). Perhaps more can be moved to the supplementary?

We thank the reviewer for raising the issue of manuscript length. In response, we have substantially shortened the manuscript.

Prior to revising, we consulted with the editorial team, who advised us to prioritize a clear and sufficiently detailed description of the methods to ensure that the work can be independently replicated. Guided by this recommendation, we focused our reductions on changes that did not compromise clarity, transparency, or completeness. Specifically, we reduced the word count through the following steps:

- *We moved methodological details not essential for understanding the present results to the Supplementary Material (e.g., sample size justification and details of the questionnaire translation).*
- *We removed redundancy between Methods and Results, while retaining brief reminders of each experiment at the beginning of the respective Results sections.*
- *We pruned the Introduction as suggested by the reviewer.*
- *We carefully edited the manuscript for unnecessarily long formulations, merging sentences where feasible. For example, the passage*
“Two QUEST staircases subsequently controlled the Michelson contrast (one staircase for each location), with the only restriction that presented contrast values had to be between 0.01 and 0.9. A Michelson contrast of 0.2 (sd = 0.3) was used as a prior (derived from piloting).”

now reads

“Two QUEST staircases (one per location) subsequently controlled the Michelson contrast within a range of 0.01–0.9, using a prior contrast of 0.2 (sd = 0.3), derived from piloting.”

In total, these changes reduced the manuscript by 1436 words (Introduction: 200; Methods: 919; Results: 294; Discussion: 5).

The revised manuscript has a total word count of 6472 words (Introduction: 768; Methods: 2577; Results: 1913; Discussion: 1214).

We believe that the revised version strikes a better balance between conciseness and the level of methodological detail required for transparency and reproducibility.

My only other concern for read-ability/access is the example at the beginning and the end of the manuscript is both long (not helping with the length of the manuscript) but also contains a lot of technical jargon outside of the expected expertise of the reader. How does my favourite team “attack” and what does it mean to “score a goal”? What makes the other player “defensive”? What is a “foul”? What is a “Dive”?

We agree with the reviewer that the real-world example at the beginning and end of the manuscript is a good opportunity to reduce the overall length and improve accessibility. We therefore revised the example to eliminate technical jargon and minimize reliance on domain-specific knowledge, while retaining its illustrative function for a broad readership. The revised version reduces the manuscript length by 73 words.

Lines 46-49:

Imagine watching a soccer match in which a shot hits the goal line, leaving it ambiguous whether the ball fully crossed the line. According to accounts of motivated perception¹⁻⁴, observers are more likely to perceive the ball as having crossed the line when the apparent goal favors their own team.

Lines 1823-1835:

In conclusion, when you watch soccer with others and shout “Goal!” while someone rooting for the other team insists that the ball did not cross the line, the disagreement may not stem from fundamentally different perceptions of the same sensory input. Rather, you might have been looking at different aspects of the event: perhaps your eyes were drawn to the goal line, while the other person was focused on the ball. Even if you both happened to focus on the same part of the scene and perceive the event in the same way, your motivation – hoping for a goal – may have biased your response to what you saw. Our findings suggest that such disagreements are not necessarily due to perceptual distortions – the ball does not appear larger⁸; the goal line does not appear closer⁹ – but rather to differences in overt attention and in how people respond to ambiguous situations, thereby identifying the behavioral pathway through which motivational states influence perceptual judgments. In short: motivation doesn’t change how our brains make sense of the sensory input – it changes where we look, and what we say.

Reviewer #2 (Remarks to the Author):

This manuscript addresses an enduring and significant debate in cognitive science regarding whether and how motivation influences perception using a series of clever experiments that combine psychophysics with eye-tracking.

In study 1, participants were asked to detect whether a numerical digit was present in a noisy stimulus across two locations, with one location linked to a higher reward for correct detection. Results showed that participants exhibited a stronger bias to report the digit as present at the high-value location compared to the low-value location, but any biases in perceptual sensitivity were explained by differences in eye-movements.

In study 2, participants were required to indicate which of two moving dots appeared faster, with a higher reward for correct responses associated with one of the dots. Here, reward magnitude was found to bias perceptual reports but not smooth pursuit eye-movements.

In studies 3 and 4, participants were presented with ambiguous visual images. Systematic differences in eye-gaze preceded perceptual shifts. When participants were asked to perceive a particular percept, their gaze was directed to the corresponding features of the stimulus. Furthermore, when instructed to direct their gaze to those features, participants were biased to report the corresponding percept.

For context, I had previously reviewed this manuscript at a different outlet. I found this version of the manuscript significantly improved. I particularly appreciated the addition of Studies 3 and 4, as they provide a convincing demonstration of how motivational effects on perceptual judgments can be mediated by differences in eye-movements. Pending some minor revisions for clarification, I would be happy to recommend publication.

We thank Reviewer 2 for the thoughtful and conceptually insightful comments. The reviewer's suggestions helped us to clarify the conceptual framework of the manuscript. We believe that these clarifications have strengthened the theoretical contribution of the paper.

1. My main comment is that manuscript isn't always clear on the criteria of what counts as motivational influences on perception. I thought the paragraph in the discussion about "what is perception?" is a valuable addition, but this more nuanced perspective seems at odds with introduction that adopts a stricter criterion.

My recommendation would be to explicitly distinguish between "direct" vs. "mediated" effects of motivation early in the manuscript, which I believe would also enhance the contribution of the current results, where the authors show *how* motivational effects on eye-movements can explain differences in perceptual judgments.

More generally, I think the field would benefit greatly from moving past the narrow question of whether motivation biases perception (the answer to which will likely depend on what one means by "motivation", "bias", and "perception", as well as the experimental context), and toward understanding the boundary conditions, pathways, and concrete perceptual differences, which I think this paper does so excellently.

We thank the reviewer for pointing out the importance of conceptual clarity. To address this, we now explicitly distinguish between direct and mediated effects of motivation on perception in the Introduction. In addition, we slightly clarified the wording in the Discussion to emphasize that perception itself is defined consistently throughout the manuscript as the construction of a representation based on sensory activity, while goal-directed actions (e.g., gaze) shape the sensory input rather than constituting perception itself. Finally, we added a brief clarifying phrase that our findings identify a behavioral pathway that links motivation to perceptual judgment.

Lines 154–157:

If motivation indeed shapes perception, then their relationship might be more nuanced than biasing the interpretation of incoming sensory information. First, motivation may not impinge directly on perception, but motivational effects may be mediated by processes such as attention and gaze.

Lines 1796–1798:

This leaves us with the question of what is perception? In our view, perception is the process of constructing an understanding of the world based on the activity of our sensors. However, this sensor activity is shaped by an inherently active, goal-directed process: We move...

Lines 1829–1833:

Our findings suggest that such disagreements are not necessarily due to perceptual distortions – the ball does not appear larger⁸; the goal line does not appear closer⁹ – but rather to differences in overt attention and in how people respond to ambiguous situations, thereby identifying the behavioral pathway through which motivational states influence perceptual judgments.

2. On a related note: Line 99: “Additionally, motivation might not necessarily bias perception, but might instead modulate perceptual sensitivity – that is, how well something is perceived rather than what is perceived.”

Wouldn't a modulation of perceptual sensitivity also be a form of bias in perception? Seeing something more clearly seems like it would count as perceptual effect.

We agree with the reviewer that changes in perceptual sensitivity would indeed constitute a perceptual effect. In our framework, bias and sensitivity are treated as two complementary key dimensions of perception. The aim of the present work was therefore to assess both dimensions, allowing us to test whether motivation affects what is perceived (bias) and/or how well it is perceived (sensitivity). We clarified this distinction in the manuscript to avoid potential confusion.

Lines 159–161:

Second, any motivational effect on perception might not necessarily manifest as a perceptual bias, but could instead affect another key perceptual dimension: sensitivity – that is, how well something is perceived rather than what is perceived.

3. Lines 419: “We used signal-detection theory to analyze the response behavior of every individual (Fig. 2A-B) and to distinguish between bias (criterion) and sensitivity (d'). Mean d'

values were $M_{high} = 0.63$ ($SD_{high} = 0.56$) and $M_{low} = 0.45$ ($SD_{low} = 0.49$) and were not larger for the high compared to the low value location, $t(59) = 1.68$, $p = 0.050$, $d = 0.22$, $BF_{10} = 0.997$ (Fig. 3B). Thus, strictly speaking, we found no difference in perceptual sensitivity between the two locations. However, without controlling for any mediating mechanisms, a tendency for better sensitivity at the high value location is apparent.”

I thought the last two sentences were written in a somewhat confusing manner, as it's not immediately clear what is meant by "strictly speaking" and "tendency". Perhaps it would be helpful to be more explicit in saying that the evidence is close but did not cross predetermined statistical threshold, and that authors will later show that any apparent motivational effect is entirely explained by eye movements?

We thank the reviewer for this helpful suggestion. We revised the paragraph to make it explicit that the difference in sensitivity did not cross the statistical threshold. And that any apparent motivational effect is entirely explained by gaze position.

Lines 1240–1246:

We used signal-detection theory to analyze response behavior (Fig. 2A-B) and to distinguish between bias (criterion) and sensitivity (d'). Mean d' values ($M_{high} = 0.63$, $SD_{high} = 0.56$, $M_{low} = 0.45$, $SD_{low} = 0.49$) were not larger for the high- compared to the low-value location, $t(59) = 1.68$, $p = 0.050$, $d = 0.22$, $BF_{10} = 0.997$ (Fig. 3B). Thus, the difference in perceptual sensitivity did not cross the predefined statistical threshold. Nevertheless, when sensitivity is compared without accounting for potential mediating mechanisms, descriptively higher d' values are observed at the high-value location.

Lines 1397–1398:

Taken together, this shows that any apparent motivational effect on perceptual sensitivity is mediated by gaze behavior.

Minor:

Figure 5B: Could the authors clarify if the fixation maps are from a single participant or the whole group?

We revised the figure caption of Figure 5B to clarify that the fixation maps show individual-level data

(B) Stimuli and fixation maps from representative individual participants when asked to bias their perception to one of two possible interpretations (red: face, seal, rabbit, young woman, 13; blue: house, donkey, duck, old woman, B). Each map depicts fixations from a single participant (participants differ across panels). Each disc represents one fixation with the disc size scaling with fixation duration.

Line 524: typo in caption of panel C: Psychometric [and] oculometric.

Thank you. Changed.

Reviewer #3 (Remarks to the Author):

This paper seeks to elucidate the source of "motivated-perception" effects by carefully considering both motivation and perception. Experiments 3 and 4, in particular provide useful evidence concerning a role of gaze behavior in effects that might otherwise be hard to interpret.

In Experiment 1, the main finding is that recognizing numbers in noise in two locations shows clear evidence that sensitivity is affected by gaze direction, which leads to biased performance. In Experiment 2 further evidence is added for separation of gaze behavior and perception. But it is really in Experiments 3 and 4 that the authors look at how gaze corresponds with observers reports of perception and also with their strategies to obtain certain perceptual information.

This is a very clear paper, which makes a clear argument. It is possible to interpret the paper as providing a means by which responses may be biased by attentional selection (gaze is overt attention), which is, of course, not surprising, but nice to see so clearly demonstrated. The stronger argument of the paper is that many "motivational" biasing effects on perception are fairly cognitive (e.g., based on attentional differences, rather than perceptual differences.) This seems like an important addition to the literature.

Signed review,
Frank Durgin

We thank the reviewer for taking the time to review our manuscript and the very positive evaluation. We appreciate the reviewer's recognition of the clarity of the manuscript and the contribution of Experiments 3 and 4 in disentangling (overt) attention and perception.